# INT-767—A Dual Farnesoid-X Receptor (FXR) and Takeda G Protein-Coupled Receptor-5 (TGR5) Agonist Improves Survival in Rats and Attenuates Intestinal Ischemia Reperfusion Injury

**DOI:** 10.3390/ijms241914881

**Published:** 2023-10-04

**Authors:** Emilio Canovai, Ricard Farré, Alison Accarie, Mara Lauriola, Gert De Hertogh, Tim Vanuytsel, Jacques Pirenne, Laurens J. Ceulemans

**Affiliations:** 1Leuven Intestinal Failure and Transplantation Center (LIFT), University Hospitals Leuven, 3000 Leuven, Belgiumtim.vanuytsel@uzleuven.be (T.V.); laurens.ceulemans@uzleuven.be (L.J.C.); 2Department of Abdominal Transplant Surgery, University Hospitals Leuven, 3000 Leuven, Belgium; 3Department of Microbiology, Immunology and Transplantation, KU Leuven, 3000 Leuven, Belgium; 4Translation Research Center for Gastrointestinal Disorders (TARGID), Department of Chronic Diseases and Metabolism (CHROMETA), KU Leuven, 3000 Leuven, Belgium; 5Laboratory of Nephrology and Renal Transplantation, Department of Microbiology, Immunology, and Transplantation, KU Leuven, 3000 Leuven, Belgium; 6Translational Cell and Tissue Research, Department of Imaging & Pathology, KU Leuven, 3000 Leuven, Belgium; 7Gastroenterology and Hepatology, University Hospitals Leuven, 3000 Leuven, Belgium; 8Department of Thoracic Surgery, University Hospitals Leuven, 3000 Leuven, Belgium; 9Laboratory of Respiratory Diseases and Thoracic Surgery (BREATHE), Department of Chronic Diseases and Metabolism (CHROMETA), KU Leuven, 3000 Leuven, Belgium

**Keywords:** intestinal ischemia reperfusion injury, Farnesoid-X receptor (FXR), Takeda G protein-coupled Receptor 5 (TGR5)

## Abstract

Intestinal ischemia is a potentially catastrophic emergency, with a high rate of morbidity and mortality. Currently, no specific pharmacological treatments are available. Previous work demonstrated that pre-treatment with obeticholic acid (OCA) protected against ischemia reperfusion injury (IRI). Recently, a more potent and water-soluble version has been synthesized: Intercept 767 (INT-767). The aim of this study was to investigate if intravenous treatment with INT-767 can improve outcomes after IRI. In a validated rat model of IRI (60 min ischemia + 60 min reperfusion), three groups were investigated (n = 6/group): (i) sham: surgery without ischemia; (ii) IRI + vehicle; and (iii) IRI + INT-767. The vehicle (0.9% NaCl) or INT-767 (10 mg/kg) were administered intravenously 15 min after start of ischemia. Endpoints were 7-day survival, serum injury markers (L-lactate and I-FABP), histology (Park–Chiu and villus length), permeability (transepithelial electrical resistance and endotoxin translocation), and cytokine expression. Untreated, IRI was uniformly lethal by provoking severe inflammation and structural damage, leading to translocation and sepsis. INT-767 treatment significantly improved survival by reducing inflammation and preserving intestinal structural integrity. This study demonstrates that treatment with INT-767 15 min after onset of intestinal ischemia significantly decreases IRI and improves survival. The ability to administer INT-767 intravenously greatly enhances its clinical potential.

## 1. Introduction

Intestinal ischemia is a frequent and potentially life-threatening condition [1]. Even when blood flow to the bowel is restored, additional injury occurs through an inflammatory process known as ischemia reperfusion injury (IRI) [2]. In this complex process, IRI causes depletion of adenosine tri-phosphate and accumulation of reactive oxygen species (ROS) precursors [3]. These mediators activate molecular and cellular components of innate immunity, leading to inflammation and cell death [4]. The subsequent disruption of the epithelial barrier causes bacterial translocation and sepsis [3,4,5]. This can eventually lead to systemic sepsis, multiorgan dysfunction syndrome, and death.

Current treatments are focused on rapid diagnosis, revascularization, and resection of necrotic bowel [1,6]. Patients also receive supportive therapy including antibiotics, anticoagulation, and aggressive fluid resuscitation [7]. However, currently there is no clinical therapy that specifically attenuates the inflammation and preserves the intestinal barrier, preventing bacterial translocation and the need for bowel resection. 

Recent studies have demonstrated the key role of the Farnesoid-X Receptor (FXR) in intestinal inflammation and intestinal barrier integrity [8]. FXR agonists have been shown to reduce intestinal inflammation by inhibiting the NF-κB pathway, resulting in reduced structural damage [9]. Our group demonstrated that obeticholic acid (OCA) (Intercept Pharmaceuticals, Morristown, NJ, USA), a semi-synthetic FXR agonist, administered before the onset of intestinal ischemia in rats significantly reduced IRI, resulting in improved survival [10]. Although our study proved the protective role of FXR activation in intestinal IRI, the limited solubility of OCA restricts its therapeutic options to a setting of oral pre-treatment (i.e., administration before onset of ischemia). While potentially useful in the controlled setting of intestinal organ donation or aortic surgery, it cannot be used to treat the more frequently occurring, unpredictable condition of acute-onset intestinal ischemia.

To overcome this, we hypothesized that the highly water-soluble, combined FXR/TGR5 agonist INT-767 (6α-ethyl-3α,7α,23-trihydroxy-24-nor-5β-cholan-23-sulfate; Intercept Pharmaceuticals, NJ, USA) administered intravenously in a therapeutic scenario 15 min after the onset of ischemia would be protective against intestinal IRI [11]. This synthetic compound is (i) highly water-soluble (making IV administration feasible), (ii) an FXR agonist ten times more potent than OCA (EC50 OCA = 0.15 ± 0.05 vs. INT-767 = 0.030 ± 0.005 [12]), and (iii) additionally is a potent agonist of the Takeda G protein-coupled Receptor-5 (TGR5) [11,12]. The latter receptor is abundantly expressed in the gastro-intestinal tract where it modulates intestinal motility and stimulates glucagon-like peptide 1 release from enteroendocrine L-cells [13,14]. TGR5 is highly expressed also in macrophages and monocytes, where its activation has shown anti-inflammatory properties by reducing expression of inflammatory cytokines following lipopolysaccharides (LPS) treatment in both pre-clinical animal and in vitro human models [15,16,17]. In rodent models of steatohepatitis and cholangiopathy, INT-767 has been shown to reduce bile toxicity and block pro-inflammatory cytokine production by macrophages [18,19].

Our aim was to test, in our validated rat model, whether intravenous INT-767 treatment protects against intestinal IRI by (1) determining the correct dosage using a pre-treatment dosing study and (2) administering it 15 min after the onset of intestinal ischemia (treatment setting).

## 2. Results

### 2.1. Experiment 1: Determination of Optimal Dosing of INT-767 Treatment

As expected, the sham rats had an intact intestinal wall, which was reflected in a high transepithelial electrical resistance (TEER) and a low fluorescein isothiocyanate-labelled dextran 20 kiloDalton (FD20) permeability (66 ± 7 Ω × cm^2^ and 16 ± 20 pmols/cm^2^) (Figure 1). IRI led to a profound damage of the intestinal wall resulting in a reduction in TEER (32 ± 7 Ω × cm^2^: *p* < 0.001) and an increased FD20 permeability (178 ± 19 pmols/cm^2^; *p* < 0.001). Administering OCA clearly led to an attenuation of the damage both in TEER (55 ± 4 Ω × cm^2^: *p* <0.001 compared to IRI) and FD20 (77 ± 19 pmols/cm^2^; *p* < 0.001 compared to IRI). 

Administering INT-767 at 5 mg/kg was not protective with regards to TEER of 33 ± 5 Ω × cm^2^ (*p* = 0.98 compared to IRI). However, INT-767 at 5 mg/kg slightly reduced the passage of FD20 (145 ± 32 pmols/cm^2^; *p* = 0.021 compared to IRI). At 10 mg/kg, INT-767 consistently increased TEER (52 ± 9 Ω × cm^2^: *p* < 0.001 compared to IRI, Figure 1a) and reduced the passage of FD20 (73 ± 27 pmols/cm^2^; *p* < 0.001 compared to IRI, Figure 1b). Based on these experiments, 10 mg/kg INT-767 was chosen as the treatment dose.

### 2.2. Experiment 2: Pharmacological Treatment of Intestinal IRI by INT-767 Administration

#### 2.2.1. INT-767 Treatment Improves 7-Day Survival

The sham-operated rats all recovered rapidly after surgery and survived (Figure 2). Intestinal IRI caused severe sepsis and led to early death in five rats (50%) (i.e., within 24 h). The remaining rats in this group died in the following 2 days due to intestinal perforations. In the INT-767 group, 3 rats out of 10 died (30%) in the first 24 h due to sepsis, and 2 additional ones (20%) died in the next two days following intestinal perforation. The surviving five rats (50%) showed normal aspects of the intestine with minimal adhesions at necropsy after one week.

#### 2.2.2. INT-767 Reduces Plasma Intestinal Injury Markers

Serum L-lactate was 1.30 ± 0.23 mmol/L in sham rats, while intestinal IRI resulted in significant increase in L-lactate (3.72 ± 1.66 mmol/L, *p* = 0.0022 compared to sham) (Figure 3a). In contrast, treatment by INT-767 resulted in plasma concentrations similar to the sham group (1.53 ± 0.45 mmol/L, *p* = 0.0049 compared to IRI). 

Likewise, I-FABP levels were very low in the plasma of the sham group (1.0 ± 0.53), but it was released in large amounts following IRI (28.77± 19.40, *p* = 0.0010 compared to sham) (Figure 3b). INT-767 treatment led to significant reduction in I-FABP to near undetectable levels (1.57 ± 2.23, *p* = 0.0015 compared to IRI). Full Western Blot can be see in Appendix A. 

#### 2.2.3. INT-767 Treatment Reduces IRI-Induced Damage to the Intestinal Wall and Ameliorates the Epithelial Barrier Function

The sham group had an intact villous structure and epithelial lining, which was reflected in a near normal Park–Chiu score (0.42 ± 0.66 (Figure 4a). IRI resulted in structural damage and thus significantly increased the Park–Chiu score to 4.89 ± 1.06 (*p* < 0.001). Rats treated with INT-767 had a markedly improved score of 1.98 ± 0.93 (*p* < 0.001 vs. IRI). The same effect was also seen for the VL, as the IRI exposed rats had shorter VL compared to the treatment group (98 ± 19 µm vs. 197 ± 58 µm, *p* = 0.044) (Figure 4b). Histological examination demonstrated significant destruction and shedding of the villi coupled with marked interstitial edema in the IRI group (Figure 4c). There was a partial attenuation of this damage in the INT-767 group.

IRI resulted in an increased intestinal permeability to ions, which was reflected by a lower TEER and an increased paracellular passage of FD20 when compared to the sham group (TEER: 13.81 ±1.99 Ω × cm^2^ vs. 48.52 ± 3.64 Ω × cm^2^ *p* < 0.001 and FD20: 75.65 ± 25.46 pmols/cm^2^ vs. 19.57 ± 12.63 pmols/cm^2^, *p* < 0.05) (Figure 5a,b). Moreover, IRI also led to increased levels of circulating LPS compared to sham rats (52.64 U/mL ± 8.8 vs. 20.33 ± 6.8 U/mL, *p* = 0.0014, Figure 5c).

Treatment with INT-767 led to an improved TEER (31.54 ± 4.50, *p* < 0.001 compared to IRI). However, the paracellular passage of FD20 was not affected by the treatment. In contrast, the transcellular permeability (mediated via endocytosis) of circulating LPS was significantly reduced by INT-767 treatment (31.49 U/mL ± 17.0, *p* = 0.028 compared to vehicle).

#### 2.2.4. INT-767 Reduces IRI-Induced Pro-Inflammatory Cytokines Expression and Upregulates Anti-Inflammatory Cytokine Expression

Intestinal damage induced by IRI caused a 24-fold increase in the expression of IL-6, which was significantly reduced in the INT-767 treated-group (*p* = 0.018) (Figure 6a, Table 1). The expression of TNF-a was 8-fold higher following IRI, which was reduced by administration of INT-767 (*p* = 0.0002) (Figure 6b, Table 1). 

IL-1b was upregulated 8-fold due to IRI but INT-767 did not significantly reduce it (*p* = 0.0742) (Figure 6c, Table 1). Similarly, IFN-γ was also upregulated 10-fold, but no difference was seen after treatment (*p* = 0.8580) (Figure 6d, Table 1). 

The expression of the anti-inflammatory cytokine IL-10 was upregulated 19-fold in the INT-767 group when comparing to the sham group (*p* = 0.0248) (Figure 6e, Table 1). Finally, expression of the Th2 cytokine IL-13 was increased 15-fold after INT-767 treatment but not statistically different compared to the IRI group (Figure 6f, Table 1).

#### 2.2.5. INT-767 Induces FXR but Not TGR5 Upregulation

The mRNA expression of FXR was reduced in the vehicle group compared to the sham group (0.36 ± 0.08 vs. 1.00 ± 0.12, *p* < 0.0001), but after INT-767 treatment, FXR expression was significantly higher compared to vehicle treatment (−0.60 ± 0.10 vs. 0.36 ± 0.08; *p* = 0.0076) (Figure 7A,C).

By contrast, the tissue protein expression of TGR5 was reduced in the sham group (*p* = 0.0293), but after INT-767 treatment no difference was observed (0.76 ± 0.24 vs. 0.70 ± 0.20; *p* = 0.9703) (Figure 7B,C).Full Western Blots can be seen in the Appendix A. 

## 3. Discussion

Intestinal ischemia is responsible for 1 out of every 1000 hospital admissions and remains a devastating condition with poor outcome and a hospital mortality of up to 80% [1,20]. This is due to a frequently insidious initial presentation often resulting in significant delay in diagnosis and treatment. Furthermore, the unique structure of the intestine can cause disastrous consequences, as the epithelium is composed of a single layer, which is the first barrier against a hostile bacterial environment. As a result, IRI can quickly lead to an increased intestinal permeability and bacterial translocation to the systemic circulation, causing sepsis and death. FXR has been identified as a key player in the pathophysiology of inflammation and intestinal barrier loss [21]. We previously showed that an oral pre-treatment with the FXR agonist OCA can protect the intestine against IRI and improve survival [10]. In the current study, we investigated the role of INT-767 and its administration 15 min after the onset of intestinal ischemia. The possibility to give this drug after onset of ischemia greatly improves its clinical potential in this indication. 

Semi-synthetic bile acid analogs represent an emerging field in therapeutics for numerous diseases such as inflammatory bowel disease (IBD), cholangitis, steatohepatitis, and metabolic syndrome [9,11,22,23]. One of the main targets is the FXR receptor, which is widely expressed throughout the entire gastrointestinal tract and in immune cells [21,24]. In a knock-out mouse study, loss of FXR expression led to increased levels of inflammation [8]. On the other hand, the FXR agonist OCA blocked the NF-κB-mediated pro-inflammatory cytokine production, including IL-1b, IL-6, and TNF-a, in animal models of IBD and intestinal IRI [9,10]. INT-767, which was synthesized from OCA by adding an organo-sulphate group, is 10 times more potent an FXR agonist, as demonstrated by a coactivator recruitment assay [11]. In a murine model of non-alcoholic steatohepatitis, INT-767 significantly reduced pro-inflammatory cytokines production by macrophages such as IL-6 and TNF-a while increasing the anti-inflammatory cytokine IL-10 [25]. These anti-inflammatory properties were confirmed in the current pre-clinical study. 

TGR5 is another bile acid receptor abundantly expressed throughout the intestinal tract, liver, macrophages, and monocytes [17,24]. It has been demonstrated to block LPS-induced NF-κB activation and activate the alpha serine/threonine-protein kinase-mammalian target of rapamycin (AKT–mTORC1) [26,27]. Our hypothesis was therefore that the addition of TGR5 agonism would provide additional protection against IRI. However, as demonstrated by our final datapoint (Figure 7), at least in this setting, this was not the case. The most important reason is most likely the timing and dosing, as we only gave a single dose of INT-767, and it was only allowed to circulate for 45 min until reperfusion. All previous data analysis in animal studies was performed on chronic exposure (multiple doses over several weeks), which would allow more time for this product to take effect [12]. An alternative explanation is that INT-767 exerts part of its effect in activating the TGR5 receptor and increases cytosolic c-AMP levels or other second messengers without affecting TGR5 expression [28].

There is a clear interaction between the initial damage (caused by ischemia) and the subsequent inflammation and increased intestinal permeability [2,29]. While a contained response is useful, the large-scale and diffuse release of pro-inflammatory cytokines will cause additional damage. Additionally, the IRI-induced ROS release causes more damage and further compromises the epithelial barrier [30]. The subsequently increased permeability is associated with the translocation of bacteria and related endotoxins, which will cause sepsis and multi-organ dysfunction [5]. This cascade was seen in our experiments and led to early sepsis and death in all untreated animals. Our present study confirmed that INT-767 attenuated this cascade and, by reducing the inflammation caused by IRI, helped maintain the structural integrity of the intestine, resulting in improved survival. 

Interestingly, while TEER (which measures epithelial ion permeability) was significantly reduced, FD20 passage (measuring the paracellular permeability for larger molecules) was not. The reasons for this apparent discrepancy are unclear, but they were also seen in our previous experiment with OCA [10]. It could be that after INT-767 treatment, limited and localized areas of necrosis (caused by the ischemia) persist, surrounded by larger preserved areas of near-normal villi. This variable damage pattern is frequently present in intestinal ischemia and could explain the discrepancy between the two parameters [31]. The lack of effect on FD20 did not impact the reduction of circulating LPS levels by INT-767. This may be because LPS tends to form complexes that are much larger than the standard FD20 molecules [32]. Moreover, recent data suggest that LPS crosses the epithelium using the transcellular rather than the paracellular pathway [33]. For example, clathrin- and lipid raft-mediated endocytosis and the chylomicron pathway have been identified to carry LPS. By contrast, it should be noted that pre-treatment with INT-767 did result in reduced FD20 passage (as seen in Figure 1b). We believe this may be due to the difference in timing (24 h pre-treatment), which would allow more time for the maximal protective effect to occur. Future experiments should explore combining INT-767 with drugs that specifically stabilize tight junction proteins to reduce intestinal paracellular permeability and potentially further improve outcomes [34]. 

FXR agonists have been studied in human trials in conditions such as primary biliary cholangitis, primary sclerosing cholangitis, and non-alcoholic steatohepatitis, which have shown encouraging results and good safety profiles [22,35]. In this study, we initially evaluated the effect of INT-767 compared to OCA, the previously described FXR agonist [10]. Both compounds seemed to offer equal protection of the intestinal wall integrity. However, the ability to administer it intravenously does provide a significant practical improvement of INT-767 over OCA in this context. 

Many drugs have been investigated in animal IRI models; however, these have mostly focused on prophylaxis, where intestinal ischemia is expected such as during aortic surgery [36]. This setting allows for administered drugs to circulate and precondition the intestine. Per definition, acute intestinal ischemia occurs suddenly and unexpectedly. Any therapy can therefore only be started after the onset. This complicates effective administration, as oral treatments are impossible due to the immediate ileus that occurs due to ischemia. Therefore, there is a paucity of practically applicable therapies that specifically attenuate the pathophysiological process of IRI. Rather, current treatment is based on rapid identification, resuscitation, and revascularization [1,37,38]. It is in this interval, between the onset of ischemia and reperfusion, that medical intervention has the greatest potential impact. Several therapies for this time interval have been described, which are aimed at reducing the effects of the resulting IRI-related damage. For, instance, antibiotics therapy, both systemic and oral, has been shown to reduce bacterial translocation and sepsis [39]. Likewise, there is evidence that administration of anticoagulation before reperfusion reduces secondary microvascular disfunction [40]. In contrast, INT-767 directly targets the underlying pathways, namely the inflammation and subsequent epithelial barrier dysfunction. Given that IRI damage is a very complex pathophysiological cascade involving several pathways, optimal therapy will have to be multimodal [7]. Therefore, one could envision administering INT-767 while patients are being prepared for revascularization. In combination with the existing therapies, this will likely reduce the need for extensive intestinal resections and improve outcomes. 

Despite the potential of this study, some limitations must be acknowledged. First, despite marked improvement in most of our endpoints, such as intestinal barrier function and inflammation, survival was only partially improved to 50%. This is most likely because IRI damage works through multiple pathways (e.g., cellular necrosis/apoptosis, oxidative stress, disturbances in the microcirculation, mitochondrial dysfunction, etc.) that one single agent cannot completely reverse. Furthermore, this model does not mimic clinical practice where surgical resections of necrotic bowel, aggressive intravenous fluid resuscitation, broad-spectrum antibiotics, and pressor support are implemented [38]. Furthermore, we used an intraperitoneal anesthetic pathway in these experiments, which we recently demonstrated increased the mortality compared to volatile anesthetics [41]. Necropsy in treated animals in our studies demonstrated that localized and limited IRI damage still occurred and resulted in perforations that were fatal despite treatment [10]. On the other hand, by choosing 60 min of ischemia and thus having a high mortality rate in the control groups, any survival effect becomes more apparent with fewer animals. Second, a 15 min treatment delay after the start of ischemia was chosen, but no other timings were explored in this study. This interval was chosen based on our previous experience with this model, mimicking a treatment delay before revascularization, as seen in clinical practice. The aims of this study were limited to investigating the correct dosing and proof of concept of the drug functioning after the start of ischemia. Future studies examining the protective effect of INT-767 at different time points should definitively be considered. 

In conclusion, the present study demonstrates that INT-767, given IV as a treatment strategy 15 min after the onset of intestinal ischemia, effectively attenuates intestinal injury and inhibits inflammation. This led to significantly lower levels of endotoxin translocation and to reduced mortality. Interestingly, contrary to our hypothesis, the INT-767 mechanism of action seems to work via FXR activation but not via the bile acid receptor TGR5. Our findings thus offer a promising avenue for further investigating the potential of INT-767 in this indication.

## 4. Materials and Methods

### 4.1. Animal Model

We utilized our validated rat model of severe intestinal IRI [10]. In short, intestinal IRI was induced by temporary occlusion of the superior mesenteric artery (60 min ischemia) in male Sprague–Dawley rats (weight 275–325 g, Janvier Labs, Saint Berthevin Cedex, France). This was performed through a midline laparotomy under intraperitoneally administered general anesthesia (combination of ketamine (Anesketin, Eurovet, Zeewolde, The Netherlands) and xylazine (Xyl-M2%, Van Miert & Dams Chemie, Berendonck, Belgium)). After 60 min of ischemia, the clamp was removed, allowing reperfusion. Sham-operated animals underwent the same procedure, including anesthesia and surgical manipulation, but without induction of intestinal ischemia. All animals were sacrificed at the end of the experiment through exsanguination under general anesthesia. Necropsy was performed on all animals. Ethical approval was granted by the local committee for animal experimentation (EC P120/2016), and animals were kept in dedicated facilities in accordance with European Union guidelines, including veterinary supervision and postoperative analgesia by buprenorphine (Vetergesic, Ecuphar, Oostkamp, The Netherlands). We used the previously described postoperative morbidity score to monitor animals [10]. Animal sample size per group was based on a power calculation extrapolated from the expected protective outcomes seen in our previous study using the same model. (See Appendix A for details).

### 4.2. Experimental Design

#### 4.2.1. Study 1: Determination of Optimal Dosing of INT-767 Treatment

To establish optimal dosing of INT-767, two concentrations were investigated in our validated and reproducible severe intestinal IRI rat model (5 and 10 mg/kg) [10] (Figure 8). We used our established pre-treatment model where the medications were administered before the start of ischemia (at 24 and 2 h prior to ischemia). This would allow direct comparison with established medication. 

The FXR dosages were based on preliminary pharmacological studies [12]. Rats were randomly assigned to five groups (n = 6/group): sham, IRI + vehicle, OCA (oral administration by gavage at 30 mg/kg-animal weight) + IRI, INT-767 (5 mg/kg) + IRI, and INT-767 (10 mg/kg) + IRI. INT-767 was administered intravenously (Figure 1). As INT-767 was water-soluble, this was dissolved in 0.9% saline. The vehicle (group 2) was 0.9% also. OCA was dissolved in 4 mL of 1% methylcellulose solution for oral administration. Based on previous pharmacokinetic studies, INT-767 was administered at 24 and 2 h prior to the onset of ischemia. All animals underwent 60 min of intestinal ischemia (except for the sham group) and were sacrificed after 60 min of reperfusion. The distal ileum was removed and mounted in an Ussing chamber to study the epithelial permeability to ions by measuring the transepithelial electrical resistance (TEER) and the paracellular permeability to fluorescein isothiocyanate-labelled dextran of 20 kilodalton (FD20, Sigma-Aldrich, Overijse, Belgium). The results for INT-767 were compared to OCA, which has demonstrated to be protective in this setting [10]. We chose TEER and FD20 as initial endpoints, as our previous work demonstrated these to provide objective and real-time data on epithelial barrier function. This dosing study revealed that INT-767 at 10 mg/kg was safe and effective and was thus used in the subsequent experiments.

#### 4.2.2. Study 2: Pharmacological Treatment of Intestinal IRI by INT-767 Administration

##### Effect of INT-767 on Survival

Three groups were studied (n = 10/group; Figure 2): 1/sham (surgery but no ischemia); 2/IRI + vehicle: 60 min ischemia + 60 min reperfusion + vehicle (0.9% NaCl at equivalent volume as INT-767); and 3/IRI + INT-767: 60 min ischemia + 60 min reperfusion + INT-767 (10 mg/kg, dissolved in a 0.9% saline solution at 2 mg/mL). The INT-767 or vehicle was administered via IV injection 15 min after start of ischemia. These time intervals were chosen based on our previous study, which showed that 60 min ischemia and reperfusion in rats provokes severe damage with limited survival, similar to 6 h of human intestinal ischemia [42]. By the same logic, the 15 min interval between onset of ischemia and start of IV treatment was chosen to represent a reversible state of damage comparable to the typical delay between onset of symptoms and diagnosis of acute intestinal infarction in a clinical setting. The animals were allowed to recover from anesthesia and observed for 7 days to document survival rate. At the end of the experiments, all surviving animals were sacrificed by exsanguination under general anesthesia (Figure 9—Experiment 2A) to perform a necropsy and collect samples (serum and bowel samples for histological and cytokine analysis).

##### Effect of INT-767 on Intestinal IRI

To study intestinal serum injury markers, epithelial barrier function, histology, and cytokine expression, six additional animals per group were sacrificed 1 h after reperfusion (Figure 9—Experiment 2B). 

At the time of sacrifice, blood was collected and centrifuged (10 min at 3500 rpm), and the resulting serum was snap-frozen and stored at −80 °C. As described previously [10], the distal ileum was divided into five segments. One segment was transported in oxygenated medium to the Ussing chamber for ex vivo functional assessment. One segment was fixed in formalin for histological analysis. The remaining three segments were snap-frozen and stored for quantitative reverse-transcriptase polymerase chain reaction (qRT-PCR) and western blot (WB).

##### Damage Biomarkers

Blood gas analysis (ABL-815, Radiometer Medical, København, Denmark) was used to test systemic L-lactate levels. WB was used to measure serum levels of intestinal fatty-acid-binding protein (I-FABP) (enterocyte damage marker) [43].

##### Histological Analysis

After formalin fixation, the ileal segments were embedded in paraffin. First, 5 µm thick sections were cut and stained with hematoxylin and eosin. A pathologist (GDH) blinded to the applied protocol and the clinical outcome analyzed four separate mucosal areas microscopically, as described elsewhere [10]. IRI damage was quantified by (1) the Park–Chiu score [44] (range: 0–8) and (2) villus length (VL), defined as the length between the tip of the villi and the mouths of the crypts (in µm). To avoid bias from segmental areas of necrosis, an average value was taken from four different sections.

##### Evaluation of the Epithelial Barrier Function

To objectively measure the epithelial barrier function, an Ussing chamber equipment was used [45]. After sacrifice, the ileal sample was divided into three segments, and a full-thickness preparation was immediately mounted on the Ussing chamber, as we previously described [10]. After a 30 min period of equilibration, the TEER was continuously measured for 120 min, and the mean value was recorded. During this process, the sample was preserved at 37 °C and oxygenated. To measure the paracellular permeability, FD20 was added to the mucosal side. After the same equilibration period as the TEER, samples were taken every 30 min. The lost volume was correct by adding buffer into the basolateral side. Serosal concentrations were subsequently measured using the microplate fluorometer FLUOstar Omega (BMG Labtech, Ortenberg, Germany). In experiment 2, both TEER and FD20 values were subsequently corrected for villus length (VL) to compensate for localized areas of denudation and necrosis. This was carried out by taking the original data and multiplying the TEER/FD20 with its corresponding VL and dividing this by the mean VL of the sham group [10].

##### Assessment of Bacterial LPS Translocation

Circulating plasma levels of LPS endotoxin, a surrogate marker for bacterial translocation, was measured using a colorimetric Limulus Amebocyte Lysate (LAL QCL1000) enzyme-linked immunosorbent assay (ELISA) (Lonza, Switzerland).

##### Quantitative Reverse-Transcription Polymerase Chain Reaction (qRT-PCR)

Relative expressions of both pro-inflammatory cytokines (interleukin (IL)-1β and IL-6, interferon (IFN)-γ, and tumor necrosis factor (TNF)-a) and anti-inflammatory cytokines (IL-10 and IL-13) as well as FXR were determined.

The snap-frozen ileal samples were homogenized in TRIzol reagent (Thermo Fisher Scientific, Dilbeek, Belgium), and the RNA was isolated using RNeasy Minikit (Qiagen, Antwerp, Belgium). Then, c-DNA was synthesized from 200 ng RNA using M-MLV transcriptase (Thermo Fisher Scientific, Dilbeek, Belgium). Lastly, the relative expressions of the cytokines were measured by qRT-PCR using a LightCycler 96 W (Roche, Vilvoorde, Belgium) with Taqman Fast Universal PCR Master Mix and Taqman Gene Expression Assays (Thermo Fisher Scientific, Dilbeek, Belgium) IL-6 (Rn01410330_m1), IL-1β (Rn00580432_m1), TNF-α, (Rn00562055), IFN-γ (Rn00594078), IL-10 (Rn00563409), and IL-13 (Rn00587615). For FXR, specific primers were designed using sequence data and nucleotide BLAST software from the National Center for Biotechnology Information database (http://www.ncbi.nlm.nih.gov/nucleotide, accessed on 15 August 2023) and were manufactured by TIB MolBiol (Berlin, Germany): FXR: 5′-cattaacaacgcgcrcacctg-3′ and 3′-ttccttagccggcaat cctg-5′. A three-step amplification was performed: 95 °C for 10 min followed by 45 cycles of amplification (95 °C for 10 s, 60 °C for 15 s, and 72 °C for 10 s), which was followed by a melting curve program. Target messenger RNA (mRNA) expression was quantified relative to the house-keeping gene GAPDH (Rn01775763_g1; Thermo Fisher Scientific, Dilbeek, Belgium) for cytokines and to Hprt1 (5′-gccaa agtggaaaagccaagt-3′and 3′-gccacatcaacaggactcttgtag-5′) for FXR using the −ΔΔCt method. Relative expression is presented as fold changes compared to the sham (i.e., mean sham value is 1).

##### Western Blot

To measure I-FABP and TGR5, stored plasma and ileal tissue samples were used, respectively. Protein concentration was determined with Bradford protein assay. Samples containing 50 µg total protein in Laemmli buffer containing β-mercaptoethanol were loaded onto a AnykDMini-ProteanTGXPrecastGel (Bio-Rad Laboratories). After electrophoresis, the proteins were transferred to a PVDF membrane using the semidry Trans-Blot Turbo Transfer system (Bio-Rad Laboratories). First, membranes with plasma samples were incubated with 0.1% Ponceau S staining solution to check equal loading. This method was chosen as this study was performed on plasma, which means there are no good housekeeping proteins. Next, the membranes were blocked for 1 h at room temperature with PBS-Tween (0.1%), which contained 5% milk powder and was incubated with the primary antibody overnight at 4 °C. The primary antibody was anti-I-FABP (21252-1-AP (1/500) Proteintech Europe, Manchester, UK) and TGR5 (PA5-23182 (1/500) Invitrogen by Thermo Fisher Scientific)). β-actin (clone AC-74 A2228 (1/30000) Abcam (Sigma by Merck)) was used as a loading control for the latter. This is followed by a 1 h incubation with the secondary antibody (anti-rabbit IgG HRP-linked antibody (7074 (1/2000) or anti-mouse IgG HRP-linked antibody (#7076 (Cell Signaling Technologies, Danvers, MA, USA)). Protein detection was performed using enhanced chemiluminescence (Pierce ECL Western Blotting Substrate) and digital detection with the ChemiDoc MP system. Quantification of relative band intensity was then performed with the associated Image Lab software (Bio-Rad Laboratories Inc, Hercules, CA, USA). The data are presented as intensity changes compared to the mean value of the sham results (i.e., mean sham value is 1).

##### Statistical Analysis

Unless otherwise mentioned, all data are shown as mean + standard deviation. In the figures, the middle line represents the mean with whiskers indicating the standard deviation. Multiple group comparisons used one-way ANOVA and post hoc Tukey test in case of normal distribution or Kruskal–Wallis with post hoc Dunn test for non-normal distribution. A Kaplan–Meier estimator was used for survival analysis (log-rank test). The distribution of results (normality) was tested using the Kolmogorov–Smirnov test. A log-rank test was used in the Kaplan–Meier curves for survival. A *p*-value < 0.05 was considered statically significant. All data sets were screened for outliers using the Robust regression and OUTlier removal (ROUT) technique using a coefficient of 5%. Statistical analysis was performed using GraphPad Prism version 9.0 (San Diego, CA, USA).

## 5. Patents

An international patent application titled “Treatment and Prevention of Intestinal Inflammatory diseases with bile acid derivative” was filed and published as WO 2020/163201.

## Figures and Tables

**Figure 1 ijms-24-14881-f001:**
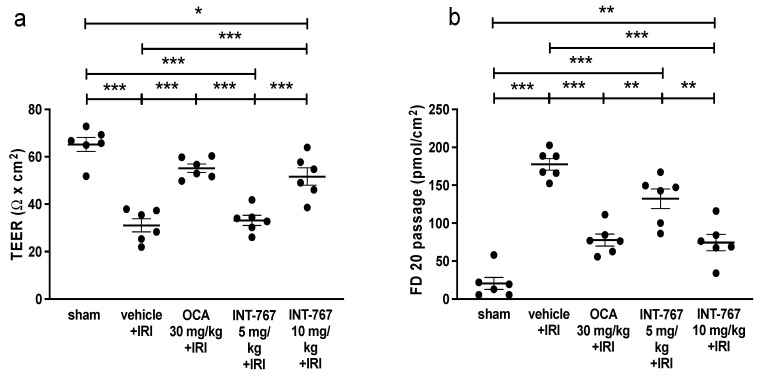
Results of Ussing chamber permeability tests when testing three compounds (OCA and two concentrations of INT-767) compared to sham and untreated rats (IRI). Permeability is expressed as (**a**) TEER and (**b**) FD20 measurements. FD20, fluorescein isothiocyanate-labelled dextran 20 kiloDalton; OCA, obeticholic acid compound; INT-767, Intercept-767; IRI, ischemia reperfusion injury; TEER, transepithelial electrical resistance; NB, vehicle (0.9% saline); *: *p* < 0.05; **: *p* < 0.01; ***: *p* < 0.001.

**Figure 2 ijms-24-14881-f002:**
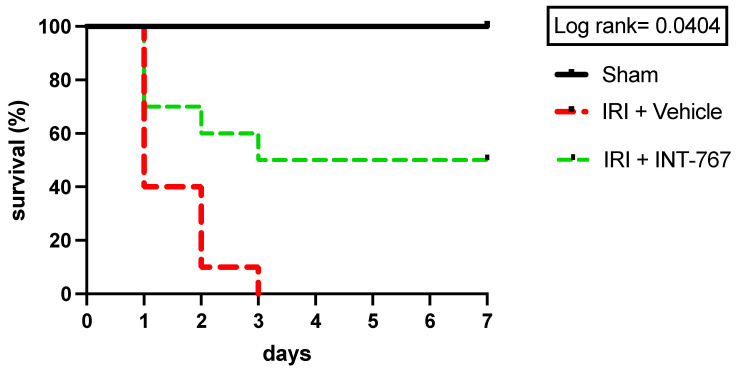
INT-767 improves the 7-day survival rate after intestinal IRI. IRI, ischemia reperfusion injury; INT-767, Intercept-767; n = 10/group; log-rank: *p* = 0.0404.

**Figure 3 ijms-24-14881-f003:**
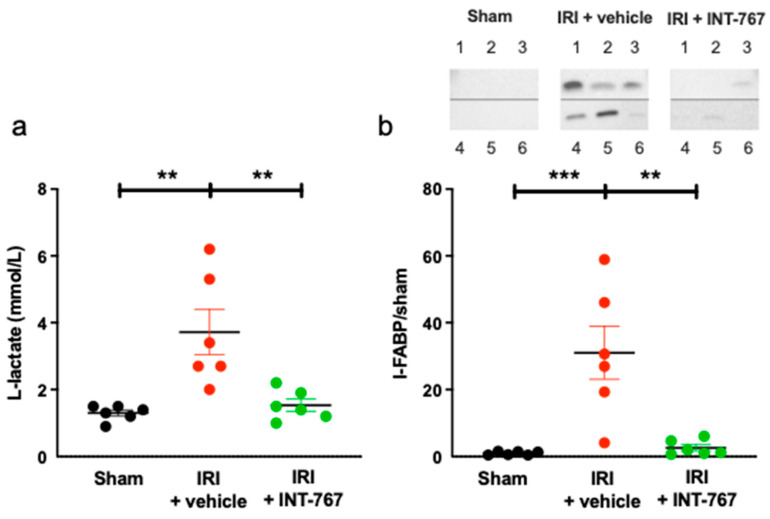
INT-767 reduces the release of intestinal IRI damage markers (**a**) L-lactate; (**b**) I-FABP (including corresponding western blot bands, NB: Top shows individual (1–3) staining, and bottom half shows individual staining (4–6). Unedited images in Appendix A). I-FABP, intestinal fatty-acid-binding protein; IRI, ischemia reperfusion injury; INT-767, Intercept-767; **: *p* < 0.01; ***: *p* < 0.001; n = 6/group.

**Figure 4 ijms-24-14881-f004:**
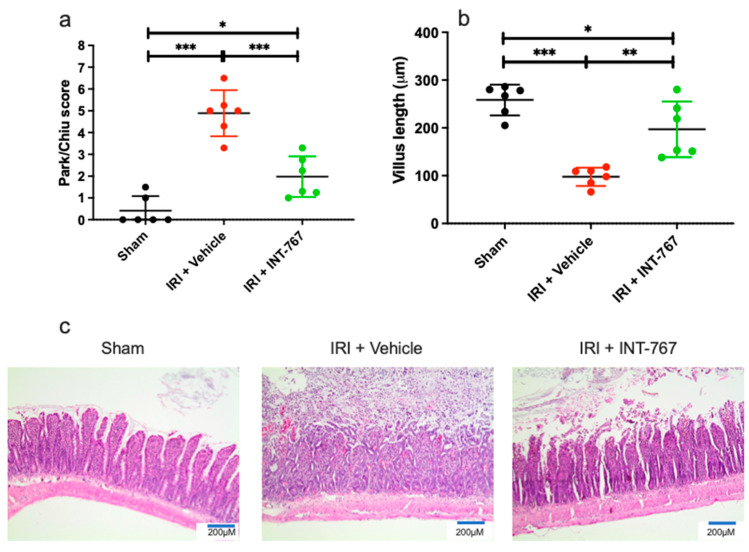
INT-767 protects the intestinal mucosal wall integrity, as shown by (**a**) Park–Chiu score; (**b**) villus length; and (**c**) representative histological illustration for each group by hematoxylin and eosin staining, magnification: ×200. IRI, ischemia reperfusion injury; INT-767, Intercept-767; *: *p* < 0.05; **: *p* < 0.01; ***: *p* < 0.001; n = 6/group.

**Figure 5 ijms-24-14881-f005:**
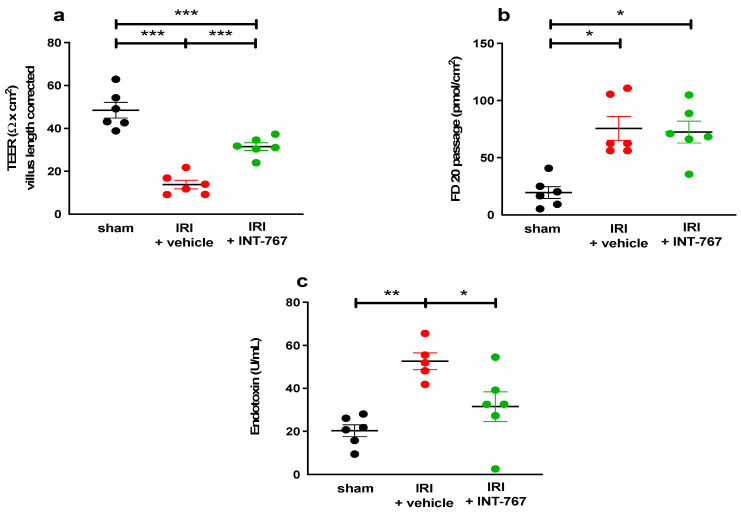
INT-767 reduces the intestinal wall permeability as demonstrated by (**a**) preservation of TEER; (**b**) reduction of FD20 levels; and (**c**) lower levels of endotoxin translocation. FD20, fluorescein isothiocyanate (FITC)-labelled 20 kiloDalton dextran; INT-767, Intercept-767; IRI, ischemia reperfusion injury; TEER, transepithelial electrical resistance; *: *p* < 0.05; **: *p* < 0.01; ***: *p* < 0.001; n = 6/group.

**Figure 6 ijms-24-14881-f006:**
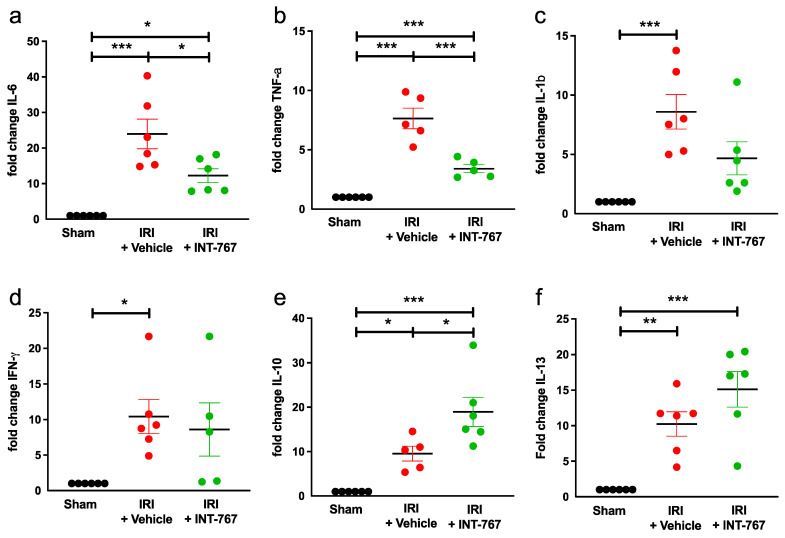
The anti-inflammatory effect of INT-767 in intestinal IRI as shown by the tissue mRNA expression of pro-inflammatory cytokines (**a**) IL-6; (**b**) TNF-a; (**c**) IL-1b; and (**d**) IFN-γ. Also shown are the mRNA expression of anti-inflammatory cytokines (**e**) IL-10 and (**f**) IL-13. IL, interleukin; IFN-γ, interferon gamma; INT-767, Intercept-767; IRI, ischemia reperfusion injury; TNF-a, tumor necrosis factor alpha; *: *p* < 0.05; **: *p* < 0.01; ***: *p* < 0.001; n = 6/group.

**Figure 7 ijms-24-14881-f007:**
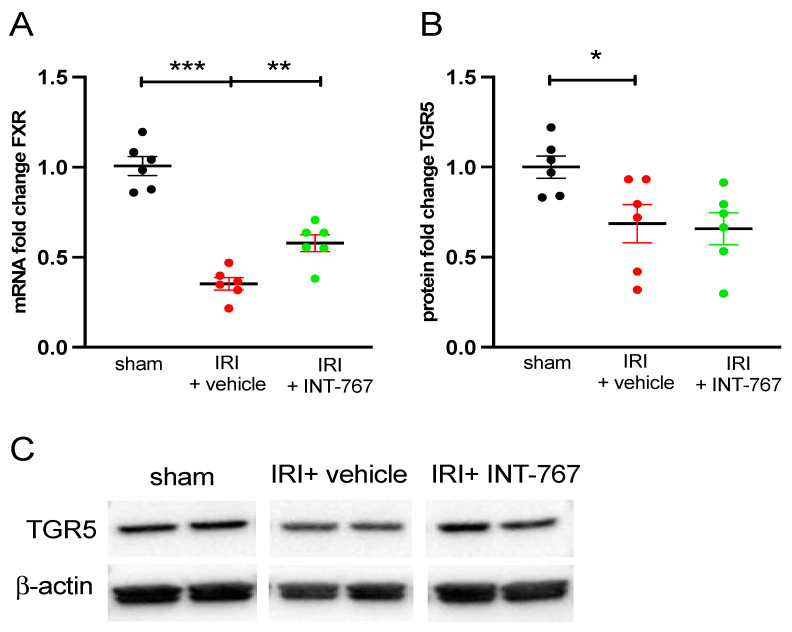
Effect of INT-767 treatment on (**A**) mRNA expression of FXR and (**B**) protein expression of TGR5. (**C**) Representative images of the western blot for TGR5 and the housekeeping protein β-actin. See unedited images in Appendix A. FXR, Farnesoid X Receptor; IRI, ischemia reperfusion injury; INT-767, Intercept-767; TGR5, Takeda G protein-coupled Receptor 5; *: *p* < 0.05; **: *p* < 0.01; ***: *p* < 0.001; n = 6/group.

**Figure 8 ijms-24-14881-f008:**
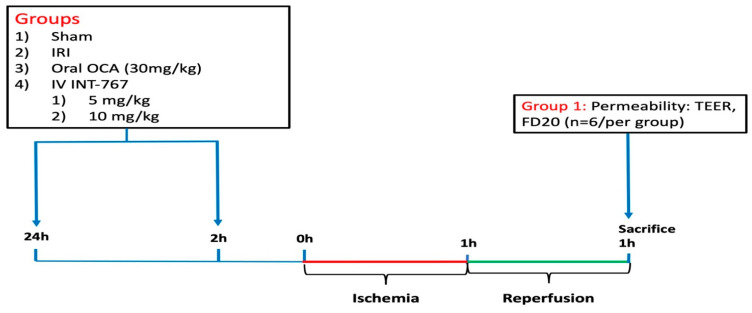
Overview of experiment 1, demonstrating the timing and dosing of the three pre-treatment groups together with the sham and control (IRI) groups. In this study, medication was administered before the start of ischemia (pre-treatment). FD20, fluorescein isothiocyanate-labelled dextran 20 kiloDalton; INT-767, Intercept-767; IRI, ischemia reperfusion injury; IV, intravenous; OCA, obeticholic acid; TEER, transepithelial electrical resistance.

**Figure 9 ijms-24-14881-f009:**
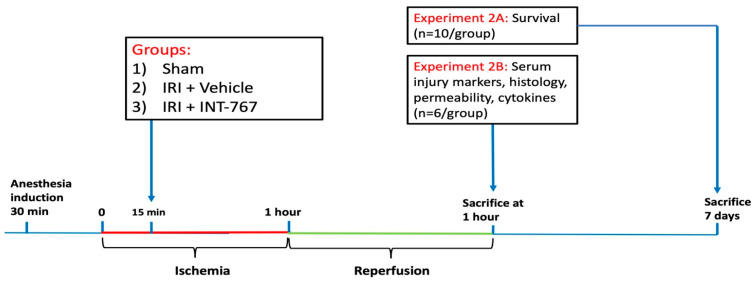
Overview of experiment 2, demonstrating the timing of administration after start of ischemia and the two study arms: the 7-day survival group (A) and the analysis after 1 h (B). IRI, ischemia reperfusion injury; INT-767, Intercept-767.

**Table 1 ijms-24-14881-t001:** Cytokine fold change data (mean ± standard deviation).

Cytokine	IRI + Vehicle	IRI + INT-767	Significance (*p*-Value)
IL-6	20.74 ± 9.31	11.2 ± 4.35	0.018
TNF-a	7.14 ± 1.74	3.26 ± 0.67	0.0002
IL-1b	7.77 ± 3.25	3.54 ± 3.11	NS (0.0742)
INT-γ	8.99 ± 5.35	8.27 ± 7.50	NS (0.8580)
IL-10	10.36 ± 3.32	16.55 ± 7.34	0.0248
IL-13	11.57 ± 3.84	17.15 ± 5.61	NS (0.1559)

IL, interleukin; IFN-γ, interferon gamma; INT-767, Intercept-767; IRI, ischemia reperfusion injury; NS, not significant; TNF-a, tumor necrosis factor alpha.

## Data Availability

Available upon reasonable request.

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
