# Peer review of "INT-767—A Dual Farnesoid-X Receptor (FXR) and Takeda G Protein-Coupled Receptor-5 (TGR5) Agonist Improves Survival in Rats and Attenuates Intestinal Ischemia Reperfusion Injury"

_ijms, 2023, doi:10.3390/ijms241914881_

Round 1

Reviewer 1 Report

This article investigated the effects of INT-767, a dual FXR and TGR5 agonist, on intestinal ischemia-reperfusion injury (IRI) in rats. The study found that treatment with INT-767 significantly improved survival, reduced inflammation, and preserved intestinal structural integrity in rats with IRI. The authors conclude that INT-767 has great clinical potential as it can be administered intravenously, significantly decreases IRI, and improves survival 15 minutes after the onset of intestinal ischemia.

This article has some innovations, including applying the INT-767 and its injection method in intestinal ischemia-reperfusion injury. However, I am mainly confused about the timing of the drug administration chosen in this study. Generally, research exploring the effect of drugs on ischemia-reperfusion injury often uses pre-treatment before ischemia or post-treatment after ischemia-reperfusion. The former mainly simulates the use of medications before surgery to explore whether drugs can reduce ischemia-reperfusion injury caused by surgical procedures, while the latter mainly refers to the use of drugs after ischemia-reperfusion injury caused by various reasons to minimize damage or death. Therefore, I do not understand the significance of administering the medicine 15 minutes after ischemia. It is recommended that the author add grouping based on the time points mentioned above to clarify the drug's therapeutic effect further.

In addition, there are multiple spelling and formatting errors in the article:

Figure 1: The resolution of Figure 1b is insufficient.

Figure 2: Please mark the statistical significance in the figure.

Figure 3: The band quality of WB is poor, and the corresponding molecular weight of the band needs to be marked, and there is no corresponding internal reference band.

Figure 4: The picture does not have a scale, or the scale is not clear.

Figure 6: There are errors in the writing format of TNF-a and IL-1b.

Reviewer 2 Report

Dear Authors, 

In this manuscript authors have described the use of INT-767 which improved the ischemia reperfusion injury (IRI) better than using Obeticholic Acid (OCA) due to its dual behavior and water solubility. However, this manuscript needed a lot of work to better understand the finding.

Major comments:

1. In figure 1 authors showed vehicle control but I am wondering how they chose one vehicle control although they have used 2 chemicals: one is water and other is DMSO soluble. Explain it, also mention the vehicle name (water or DMSO in figure legend)

2. In the method section authors mentioned gavage but they did not mention the volume so how they did this; on the other hand, they mentioned mg/kg. it is supposed to be mg/body weight.

3. showed the timeline in Figure 1 instead of figure 7. Why is the time discrepancy between text and timeline?

4. In figure 2b, what are the upper and lower panel in WB? Please showed the respective endogenous control.

5. Why the figure 5b is different from the figure 1b. explain it.

6. In figure 6 authors showed m-RNA data but I would like to see the protein data as well. Without protein data it is hard to say that INT-767 is functionally important.

7. In line 205-207, they mentioned INT-767 treatment, 15 min after the onset of intestinal ischemia but I did not see in the result section so please mention it at the first place where you are using this timeline (result section)

8. authors talked about TGR5 and NF-KB but they did not study any of them at least they should showed the data for TGR5 by western.

Minor comments:

1.    Please pay attention to the font, line 93 and 96 are different from the rest.

2.    Please incorporate the scale bar on histology data.

3.    Line 163 to 173 is the repetition of line 152 to 162.

4.    Correct the figure labeling keep it same in text and figure either lower or capital letter.

5.    Please write the discussion according to your finding.

Reviewer 3 Report

The study provides preliminary results supporting the efficacy of INT-767 in reducing intestinal damage in IRI and is an addition to a similar work presented by the research team using OCA. The results of this study are relevant since they do not focus on prophylaxis but rather on the treatment of IRI 15 minutes after the injury.  The manuscript is well-written, and the claims are backed by relevant data. This research adds value to the field by providing proof of the concept of a potent water-soluble INT-767 for IRI.

The following suggestions could improve the manuscript further before publishing.

1.     The authors have cited their previous work (Ceulemans LJ et al, 2017) testing the effect of FXR-agonist obeticholic acid (OCA) on the attenuation of intestinal ischemia-reperfusion injury. Since INT-767 and OCA are both semisynthetic bile acids, a section (in the discussion) in the manuscript to briefly compare the efficacy of these compounds would add more value. Is INT-767 better or at par in efficacy compared with OCA?

2.     Can the authors describe why only two dosages(5 and 10mgkg were used to test the efficacy), was there any specific reason for not conducting a dose-escalating study?

3.     Additionally, a 10% 7-day survival was reported in Ceulemans LJ et al, 2017 for IRI animals, while in the current study, no animal with IRI (+vehicle) survived beyond 3 days. Can the authors provide a reason why the survival rate was reduced in the current study? 

4.     INT-767 treatment has no impact on FD20 passage indicating that paracellular permeability is increased in INT-767-treated IRI rats. Could the authors discuss the negative consequences of increased paracellular permeability and suggest some future studies that would aim at reducing the paracellular movement for better long-term outcomes?

5.     Line48: Reactive instead of radical oxygen species

6.     Line 75: Please mention the EC50 of the molecules used.

7.     Fig 1b: Image quality could be better.

8.     116-122: To maintain consistency and easy interpretation it would be great to have the number of mice also mentioned along with the percentages.

9.     Fig 3B: the western blot images should be labeled.

10.  Histological illustrations in Figure 4C are blurry.  Also, the font size of the Asterix used in the figure should be consistent across the manuscript.

11.  The values mentioned in the text of the manuscript break the flow of information. It would be good if authors could mention the fold change of readouts in treated vs non-treated groups and mention the exact values in a separate table. This will allow the user to look at all the values tested at a glance. 

Round 2

Reviewer 2 Report

The Authors have responded to all my criticisms as best they can and I'm happy with them. There are some minor changes below, look at them and consider, if possible, otherwise it's fine by me.

1.     In figure 1 authors showed vehicle control but I am wondering how they chose one vehicle control although they have used 2 chemicals: one is water and other is DMSO soluble. Explain it, also mention the vehicle name (water or DMSO in figure legend)

Thanks for pointing this unclarity out to us. For INT-767 we used 0.9% saline as our solvent and this was also used in the vehicle in group 2 (vehicle + IRI). In group 3, we used 1% Methylcellulose as our solvent for the oral gavage. Both of were added to text and figure legends.

 I do not see Methylcellulose is in figure1 legend.

We decided to use the IV 0.9% saline in our vehicle as the main aim of the study was to verify the effect of IV INT-767. The oral OCA group was added based to see if the effect-size of IV treatment was comparable to the historic protective data from this compound. We therefore did not decide to use the other control as this would repeat the data already available from the previous study (Ceulemans et al 2017). We felt this would require unnecessary duplication of study groups with ethical implications under RRR principle in study animal welfare.

Authors are claiming that using of vehicle control for OCA is unnecessary duplication of study groups with ethical implications under RRR principle in study animal welfare because of previously published data (Ceulemans et al 2017) which have control, but I do not see control in this paper too. Therefore, I do not understand how they are claiming which even not exit (Methylcellulose name is not in 2017 paper). 

2.     showed the timeline in Figure 1 instead of figure 7. Why is the time discrepancy between text and timeline?

Apologies for this typing error. In the pre-treatment, the compound was administered 24 and 2 hours (not 4 as it mentions in the text). The figure 7 is correct, this was amended in the text.

Please show this timeline in Figure 1, so that reader understand, showing it in the method section is good, if it starts at before your result but after result, it is not sound scientifically. Also correct the Figure name, it is 8 not 9 (see the current manuscript). Correct the figure labelling including sublabels.

1.